# Dynamic changes of lung sRAGE in mice with chronic obstructive pulmonary disease induced by cigarette smoke exposure

Yue He[1‡], Hongyu Liang[1‡], Xiaohui Yang[1], Fengyun Hao[2], Kai Huang[3], Qiang Wang[1]*

1 Department of Respiratory and Critical Care Medicine, The Affiliated Hospital of Qingdao University, Qingdao, China, 2 Department of Pathology, The Affiliated Hospital of Qingdao University, Qingdao, China, 3 Department of Radiology, The Affiliated Hospital of Qingdao University, Qingdao, China

‡ YH and HL contributed to the work equally and should be regarded as co-first authors.
* qiangwang1111@163.com

## Abstract

### Objective

To study the changes of lung function, pathophysiology, inflammatory cytokines and related inflammatory responses in COPD mouse model, and to analyze the role of sRAGE in the pathogenesis of COPD induced by cigarette smoke (CS) exposure in mice.

### Methods

24 healthy male C57BL/6J mice aged 6 to 8 weeks were randomly divided into Smoke-Exposed (SE) group and Control group. The mice in SE group were exposed to 7 time points at 3, 7, 15, 30, 60, 90 and 120 days, while mice in control group were exposed to fresh room air, with 3 mice in each group. Lung function of mice was detected at different exposure time points, and the lung tissue sections were stained with HE to observe the lung histopathological changes of mice in each group, and the lung tissue morphological quantitative analysis was performed to evaluate the degree of emphysema. The content of inflammatory cytokines including IL-1β, IL-6 and TNF-α in the supernatant of BALF was detected by ELISA to evaluate the pulmonary inflammation of mice. The expression of sRAGE in BALF supernatant was detected by ELISA. BALF cell precipitates were classified and counted under light microscope.

### Results

After 90 days of exposure to cigarette smoke, the lung function of mice was significantly reduced, emphysema appeared significantly, and the expression of inflammatory cells and inflammatory cytokines in BALF was significantly increased (all P<0.05). sRAGE increased significantly in the early stage of CS exposure (7–15 days) compared with the control group, and the number of macrophages and levels of inflammatory cytokines in BALF also increased temporarily (P<0.05). With the gradual exposure of CS, sRAGE expression gradually decreased, and was significantly reduced after COPD formation compared with the control group.

**Data Availability Statement:** All relevant data are within the paper and its Supporting Information files.

**Funding:** This work was supported by the Wu Jieping Medical Foundation, (320.6750.19092-6). The author who received the funding: Qiang Wang Funder: Wu Jieping https://www.wjpmf.org.cn/ No funders play any role in the study design, data collection and analysis, decision to publish, or preparation of the manuscript.

**Competing interests:** The authors have declared that no competing interests exist.

## Conclusion

In the process of the occurrence and development of chronic obstructive pulmonary disease induced by cigarette smoke exposure, the level of sRAGE in bronchoalveolar lavage fluid showed a dynamic change of first increase and then decrease. The expression of sRAGE increased in the early stage of smoke exposure and played a transient pro-inflammatory role. With long-term exposure to cigarette smoke, the inflammatory response is gradually aggravated in lung, and the expression of sRAGE is significantly decreased, and its reduction degree is closely related to the degree of reduced lung function and inflammation.

## 1 Introduction

Chronic Obstructive Pulmonary Disease (COPD) has one of the highest morbidity and mortality rates globally. Chronic inflammation is the main feature of COPD, and its development is associated with a variety of cells and cytokines [1]. In recent years, several studies have found that the advanced glycation end products (AGEs) with its cell-bound receptor RAGE are widely involved in the development of COPD [2]. As one of the soluble forms of RAGE, sRAGE can competitively bind to RAGE ligands and inhibit the pro-inflammatory effects of the AGE-RAGE/sRAGE axis [3]. The sRAGE plays a key role in the pathophysiological processes of COPD, but there is a lack of consistency in the studies of its association with smoking. Several studies have shown that the expression of sRAGE is significantly lower in the peripheral blood of COPD patients compared to healthy controls [4–8]. However, there is still controversy about how smoking affects changes in sRAGE levels [5, 7–9]. The lack of consensus among different studies on the effect of cigarette smoking on sRAGE levels limits the diagnostic value of sRAGE as a biomarker of COPD.

Cigarette smoking is the most important risk factor for COPD, and cigarette smoke can mediate the chronic inflammatory process in COPD through the AGEs-RAGE/sRAGE axis [10]. In order to clarify the change rule of sRAGE during the development of COPD, we established the COPD mouse model induced by cigarette smoke inhalation, and dynamically observed the changes in the expression of sRAGE and the levels of various inflammatory cytokines in the BALF of mice. In addition, we tested lung function indexes and lung histopathological analysis in mice in order to investigate the relationship between sRAGE and smoking-induced changes in lung lesions. Our study provides some theoretical basis for the clinical application of sRAGE as a biomarker of COPD and the development of new therapeutic approaches.

## 2 Materials and methods

### 2.1 Experimental animal

Male C57BL/6J mice, 6~8 weeks old, SPF grade, weighing about 20 g/pupil, were purchased from Jinan Pengyue Laboratory Animal Breeding Co. Ltd. and housed in the SPF-grade animal room of the Animal Experiment Center of the Science and Education Building of the Affiliated Hospital of Qingdao University. Each group of mice was housed in conventional animal facilities with free access to food and water. All the experiments were conducted in accordance with the "Guide for the Care and Use of Laboratory Animals" issued by the National Institutes of Health of the United States of America and approved by the Experimental Animal Ethics Committee of the Affiliated Hospital of Qingdao University (Approval number: AHQU-MAL20210723).

## 2.2 Main equipment and reagents

Tobacco was purchased from Qingdao Yi Zhong Tobacco Group Co., Ltd. under the brand name of Hardman (tobacco tar 10mg/cigarette, nicotine 1mg/cigarette, carbon monoxide 11mg/cigarette), Mouse FlexiVent Lung Function Instrument (Beijing Emuka Biotechnology Co., Ltd.), Intravenous Needle (Straight, 22G, Weihai Jieri Medical Products Co., Ltd.), Full-wavelength Multifunctional Enzyme Marker (TECAN, Switzerland), Diff-Quick Stain (Solebo, China), HE Staining Kit (Biyuntian, China), IL-1β Mouse Elisa Kit (ABclonal, USA), IL-6 Mouse Elisa Kit (ABclonal, USA), TNF-α Mouse Elisa Kit (ABclonal, USA), sRAGE mouse Elisa kit (Kolu, China).

## 2.3 Experimental methods

**2.3.1 Establishment of mouse model.** 24 healthy male C57BL/6J mice were randomly divided into a cigarette smoke exposure group and a control group, with a total of 8 groups and 3 mice in each group. There are 7 groups exposed to cigarette smoke, namely tobacco smoke exposure for 3, 7, 15, 30, 60, 90, and 120 days, and 1 control group without tobacco exposure as a blank control. Referring to a previously published method for establishing the COPD mouse model [11], mice were placed in a homemade transparent Plexiglas fume box (80×65×50 cm) with a 1.5 cm diameter ventilation hole on the top of the box, and a Plexiglas divider (1 cm diameter circular ventilation holes on the divider, row density: 1/ 10 cm 2) was disposed at 1/2 height of the box to divide the box into upper and lower parts. The mice were placed on the partition and 10 cigarettes were lit in the lower 1/2 of the glass box, and fresh cigarette smoke was inhaled continuously for 60 minutes each time. Mice in each group were exposed to cigarette smoke twice daily, with an interval of >4 hours between smoking sessions, and were kept normally outside of the smoking sessions. Control mice were exposed to fresh indoor air without any treatment. Emphysema occurs and develops as tobacco smoke exposure continues. The successful construction of a COPD mouse model was verified by observing lung function, histopathological changes and detecting inflammatory cytokines. All of the above animal use and related experimental operations were performed under the approval of the Animal Management Committee of the Affiliated Hospital of Qingdao University.

**2.3.2 Observation of general symptoms.** Observe the mice's respiratory condition (with or without coughing, wheezing, etc.), food intake, activity level, hair and body weight changes. Weigh the mice before the start of the experiment and every 15 days during the experiment, and observe the difference in dynamic changes in body weight growth of each group of mice.

**2.3.3 Measurement of lung function in mice.** Mice were anesthetized intraperitoneally with sodium pentobarbital (50mg/kg body mass) at different time points of cigarette smoke exposure, and after the disappearance of spontaneous respiration and pain reflexes, the trachea was separated and exposed, and a "V"-shaped incision was made between the cartilaginous rings, and a metal cannula was inserted into the trachea in the centripetal direction and connected to the lung function detector (Fig 1). The mice were recorded with 0.1-second forceful expiratory volume (FEV0.1), forceful lung volume (FVC), peak expiratory flow rate (PEF), and central airway resistance (Rn). Data were analyzed using FlexiWare v 7.2 software, and all steps were performed in strict accordance with the manufacturer's guidelines.

**2.3.4 Bronchoalveolar lavage.** Bronchoalveolar lavage was performed immediately after measurement of lung function. Thread the proximal end of the trachea of mice, dispose of the 22G intravenous needle cannula in the "V"-shaped incision, secure it with silk thread, inject 0.5mL of pre-cooled PBS buffer, gently press the sternum and slowly pump back under negative pressure. Carry out the next lavage every 30 seconds and repeat the lavage 3 times for each

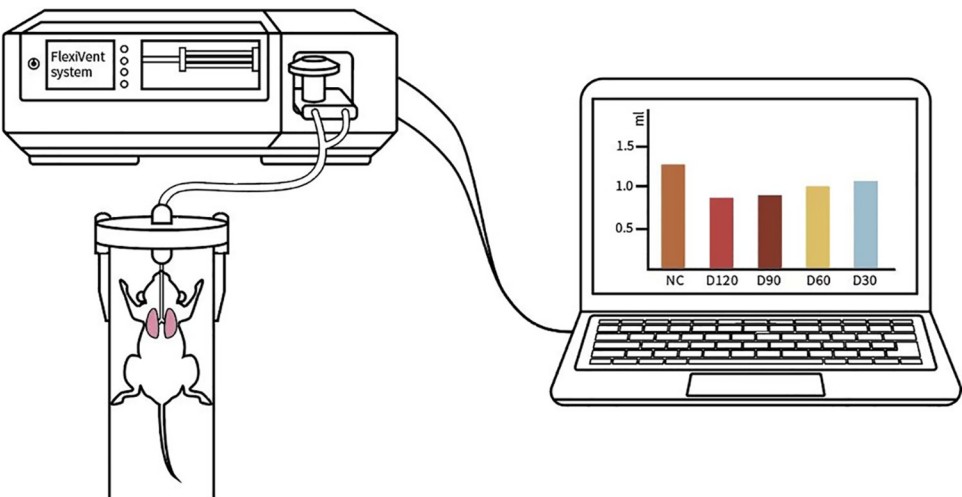

**Fig 1. Schematic diagram of FlexiVent lung function detection.**

mouse, with the recovery rate more than 80%. After bronchoalveolar lavage, mice were then sacrificed through pentobarbital injection of 500 mg/kg. The collected bronchoalveolar lavage fluid was centrifuged at 1500 r/min at 4°C for 10 min and the supernatant suspensions were frozen for subsequent testing. Moreover, the cell precipitate at the bottom of BALF was collected for Diff-Quik staining. The stained cell precipitation was classified and counted under light microscope. A total of 400 cells were counted under high magnification field of view, and the number of neutrophils, macrophages and lymphocytes among them was counted.

**2.3.5 Lung histopathologic examination.**   Mouse lung tissues were removed after bronchoalveolar lavage, and the left lung tissues of mice were sheared into small tissue blocks and immediately placed in 4% paraformaldehyde solution and fixed at room temperature for 48 hours. The lung tissues were embedded in paraffin, sectioned, stained by HE, and sealed. The histopathological morphology and structural changes of the sections were observed under light microscope.

**2.3.6 Morphological observation of lung tissue.**   Tissue sections were imaged 200 times by Case Viewer software. Three images were randomly selected from each section to measure the following indicators to evaluate the degree of emphysema. Using Image-Pro Plus software, measure mean linear intercept (MLI), mean alveolar number (MAN) and the ratio of parenchymal area to total lung area (PAA) for each image.

**2.3.7 Enzyme-linked immunosorbent assay (ELISA).**   The standard and sample wells were set up on the enzyme label plate, and after the steps of closure, sample addition, and antibody incubation, the absorbance OD value of each well was detected at 450 nm on the enzyme label instrument. A standard curve was drawn based on the OD450 value as the horizontal coordinate and the absorbance of the gradient-diluted standards of each protein as the vertical coordinate. The contents of sRAGE as well as the inflammatory cytokines IL-1β, IL-6 and TNF-α in the supernatant of BALF of mice in each group were calculated separately using the standard curve.

## 2.4 Statistical analysis

Experimental data were statistically analyzed using SPSS 26.0 software, and Graph Pad Prism 9.0 software was used for image drawing. Independent Samples t-test was used for comparison

between two groups, One-way ANOVA was used for comparisons between multiple groups, LSD-t test was used for two-way comparisons between groups, and Pearson correlation analysis was used for correlation analyses. $p < 0.05$ was considered as statistically significant difference.

## 3. Result

### 3.1 Observation of general symptoms in mice

The control group mice had clean and shiny coats, they were active and breathed steadily. However, the fur of the SE group mice gradually turned dry and yellow, they have reduced activity and occasionally cough and sneezeactivity. The weight of control group mice increased significantly but the weight of SE group mice was significantly delayed. The body weight of control group mice was significantly different from the SE group mice after 60 days of CS exposure (P<0.01) (Fig 2).

### 3.2 Effect of cigarette smoke on lung function in mice

The results of lung function in mice showed that the $FEV_{0.1}$, FVC and $FEV_{0.1}$/FVC of 90-D and 120-D SE group mice were significantly lower than those in the control group (Fig 3A–3C) (P<0.05). The peak expiratory flow (PEF) of 120-D SE group mice was significantly lower than that in the control group (Fig 3D) (P<0.001). In addition, the central airway

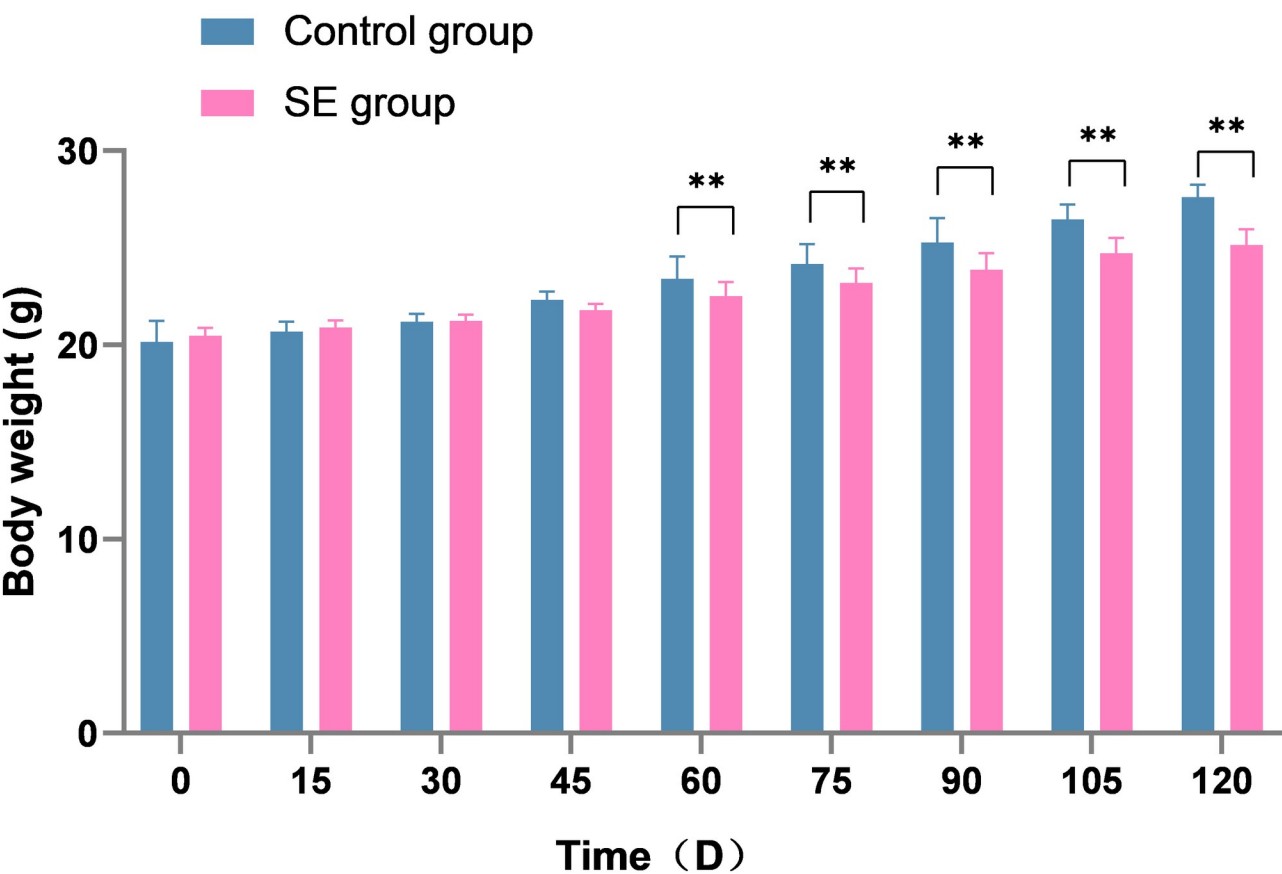

**Fig 2. Comparison of body weight gain between control group and cigarette smoke exposed group mice.** Note: ** compared with the control group: P<0.01.

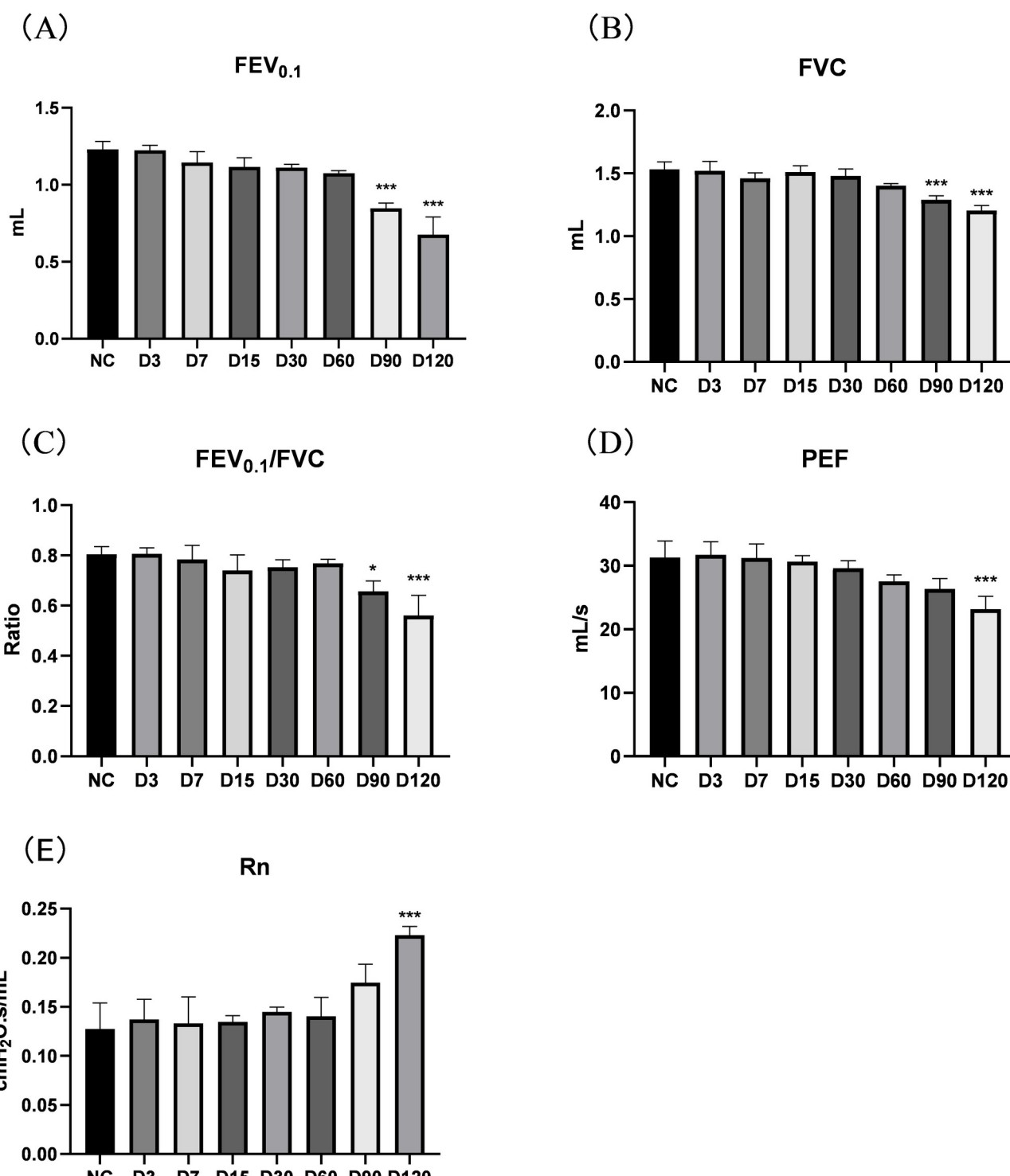

**Fig 3. Changes in lung function of mice exposed to continuous inhalation of CS.** (A) FEV0.1: forced expiratory volume in 0.1 seconds; (B) FVC: forced lung capacity; (C) FEV0.1/FVC: 0.1 second rate; (D) PEF: Peak expiratory flow rate; (E) Rn: Central airway resistance. Note: There is a difference compared to the control group, *: $p < 0.05$; * * *: $P < 0.001$.

resistance (Rn) of 120-D SE group mices was significantly higher than that in the control group (Fig 3E) (P<0.001). These results showed that cigarette smoke exposure could induce obstructive ventilatory dysfunction characterized by emphysema in mice, and lung function decreased with prolonged cigarette smoke exposure.

### 3.3 Effect of cigarette smoke on lung histopathological morphology in mice

The lung tissue structure of the control group mices were normal, with the specific characteristics of neatly arranged bronchial cilia, no congestion in the lung tissue, no inflammatory cell infiltration around the trachea, uniform size and complete structure of the alveoli, and no expansion or fusion. As cigarette smoke exposure continued, significant emphysema and inflammatory infiltration appeared in the D90 and D120 SE groups, with specific characteristics of extensive alveolar expansion and fusion, alveolar wall rupture, ribbon-like changes, enlarged alveolar cavities, infiltration of macrophages and lymphocytes and capillary dilation and congestion. After 90 days of smoke exposure, mice developed emphysema (Fig 4A). The severity of emphysema was determined by analyzing the MLI, MAN, and PAA in each group (Fig 4B–4D). After 60 days of CS exposure, the MAN was significantly less than that of the control group (P<0.05), but there was no significant difference in MIL and PAA (P>0.05).

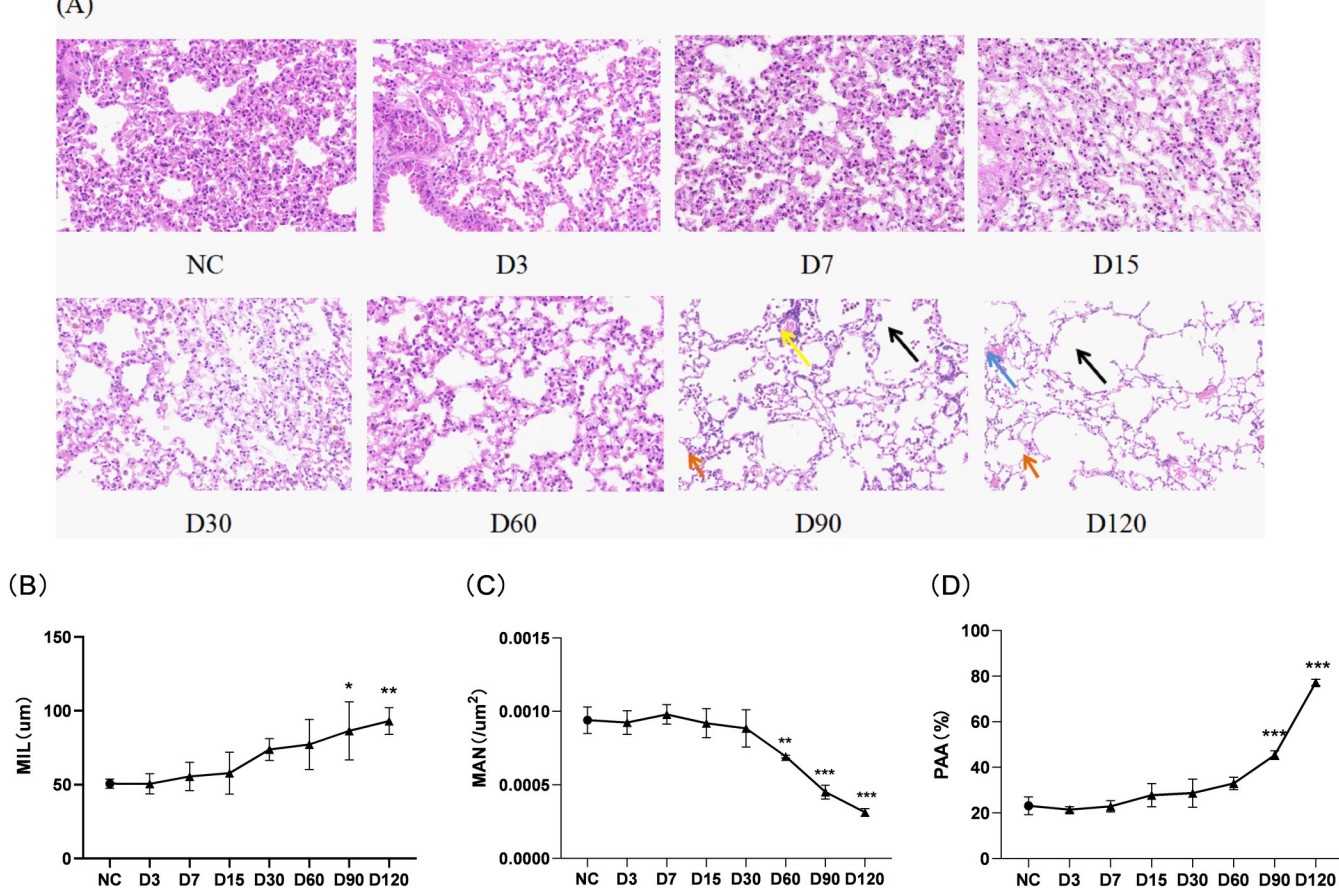

**Fig 4. Pathological changes in lung tissue of mice exposed to continuous inhalation of CS.** (A) HE staining of lung tissue sections (x 200 times); (B) Mean linear intercept (MIL); (C) Mean alveolar number (MAN); (D) The ratio of parenchymal area to total lung area (PAA). (Black arrow: alveolar wall rupture, alveolar dilation and fusion; orange arrow: inflammatory cell infiltration; blue arrow: capillary congestion; yellow arrow: lymphocytic lesion). Note: There is a difference compared to the control group, *: p<0.05; **: P<0.01; ***: P<0.001.

The MIL and PAA in the D90 and D120 groups were significantly higher than those in the control group (P<0.05), while the MAN was significantly decreased (P<0.001). The above results indicated that after 90 days of CS exposure, the lung histopathological morphology in mice showed obvious emphysema-like changes, and the degree of emphysema continued to increase with the extension of cigarette smoke exposure.

### 3.4 Effect of cigarette smoke on the number of inflammatory cells in BALF in mice

Sort and count the stained cells under light microscopy to calculate the percentage of three types of inflammatory cells (Fig 5A). The results showed that the total number of inflammatory cells in BALF of SE group mices increased progressively. The total number of inflammatory cells increased significantly after 7 days CS exposure, and the total number in the D120 SE group mices increased approximately 4-fold compared with the control group (P<0.001, Fig 5B). There were a small number of macrophages in the BALF of mice under baseline conditions. Compared with the control group, the number of macrophages in the D7 SE group mices increased significantly, and began to decrease after reaching the first peak on the 15th day, and increased nearly 3 times after 120 days of CS exposure (P<0.001, Fig 5C). Neutrophils were almost absent in the control group mice BALF. After exposure to tobacco smoke, it showed the highest increase in quantity (P<0.001, Fig 5D). There was no significant change in lymphocytes at the early stage of smoke exposure (15 days ago). It showed a progressive increase after 60 days CS exposure (P<0.05, Fig 5E). The number of all three types of cells showed a progressive increase with prolonged cigarette smoke exposure, suggesting that macrophages, neutrophils, and lymphocytes all play important roles in cigarette smoke-induced lung tissue injury and inflammatory response.

### 3.5 Effect of cigarette smoke on the expression of inflammatory cytokines in BALF of mice

IL-1β, IL-6, and TNF-α are important inflammatory cytokines in the development of COPD. The three inflammatory cytokines in the BALF of SE group mice showed a transient increase in the early stage, followed by a decrease to normal (Fig 6). As the smoke exposure time prolonged, after 60 days, the content of IL-1β and IL-6 significantly increased compared with the control group (P<0.05); TNF-α also significantly increased compared with the control group after 90 days (P<0.001). With long-term exposure to CS, the levels of the three inflammatory cytokines all showed a progressive increase and the inflammatory response in the lung gradually worsened.

### 3.6 Effect of cigarette smoke on sRAGE expression in mouse BALF

The content of sRAGE in mouse BALF began to increase after 3 days of CS exposure, which was significantly higher than the control group at 7–15 days (P<0.05), and then gradually decreased. The content of sRAGE in CS exposure was significantly lower than the control group after 90 days CS exposure (P<0.01) (Fig 7). The results suggest that the sRAGE level briefly increased in the early stage of cigarette smoke exposure (7–15 days) and then with the continuous exposure of CS, the content of sRAGE in BALF showed a further downward trend.

### 3.7 Correlation between sRAGE levels and pulmonary function decline and inflammatory response

Pearson correlation analysis was used to assess whether there is a potential link between sRAGE and lung function and lung inflammatory response. After 120 days CS exposure, there

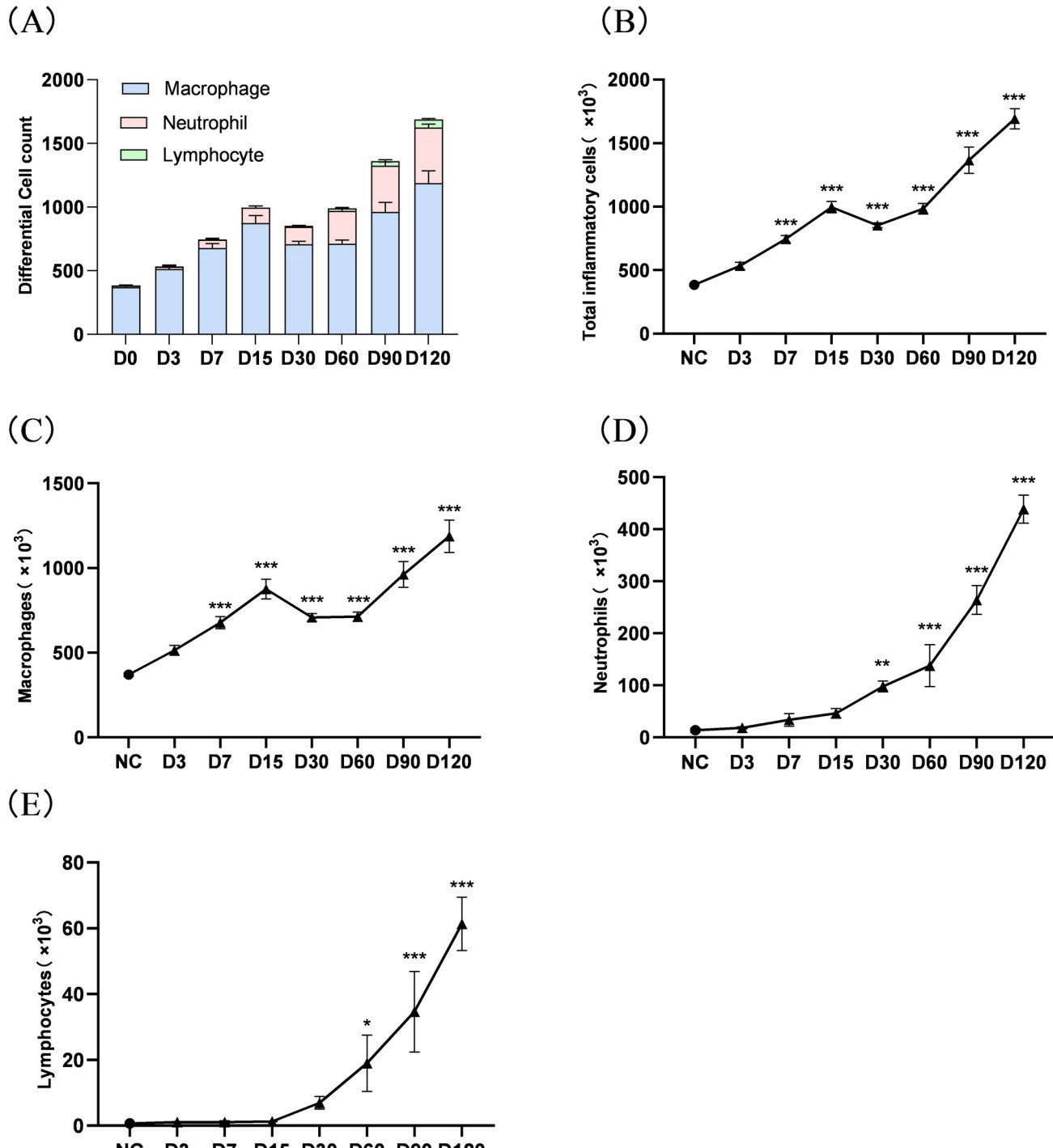

**Fig 5. Changes in inflammatory cell count in BALF of mice exposed to continuous CS inhalation.** (A) The proportion of macrophages, neutrophils, and lymphocytes to the total number of cells in the lavage fluid in each group; (B) The total number of inflammatory cells in each group of BALF; Number of macrophages, neutrophils, and lymphocytes in each group of BALF (C-E). Note: There is a difference compared to the control group, *: p<0.05; * *: P<0.01; * * *: P<0.001.

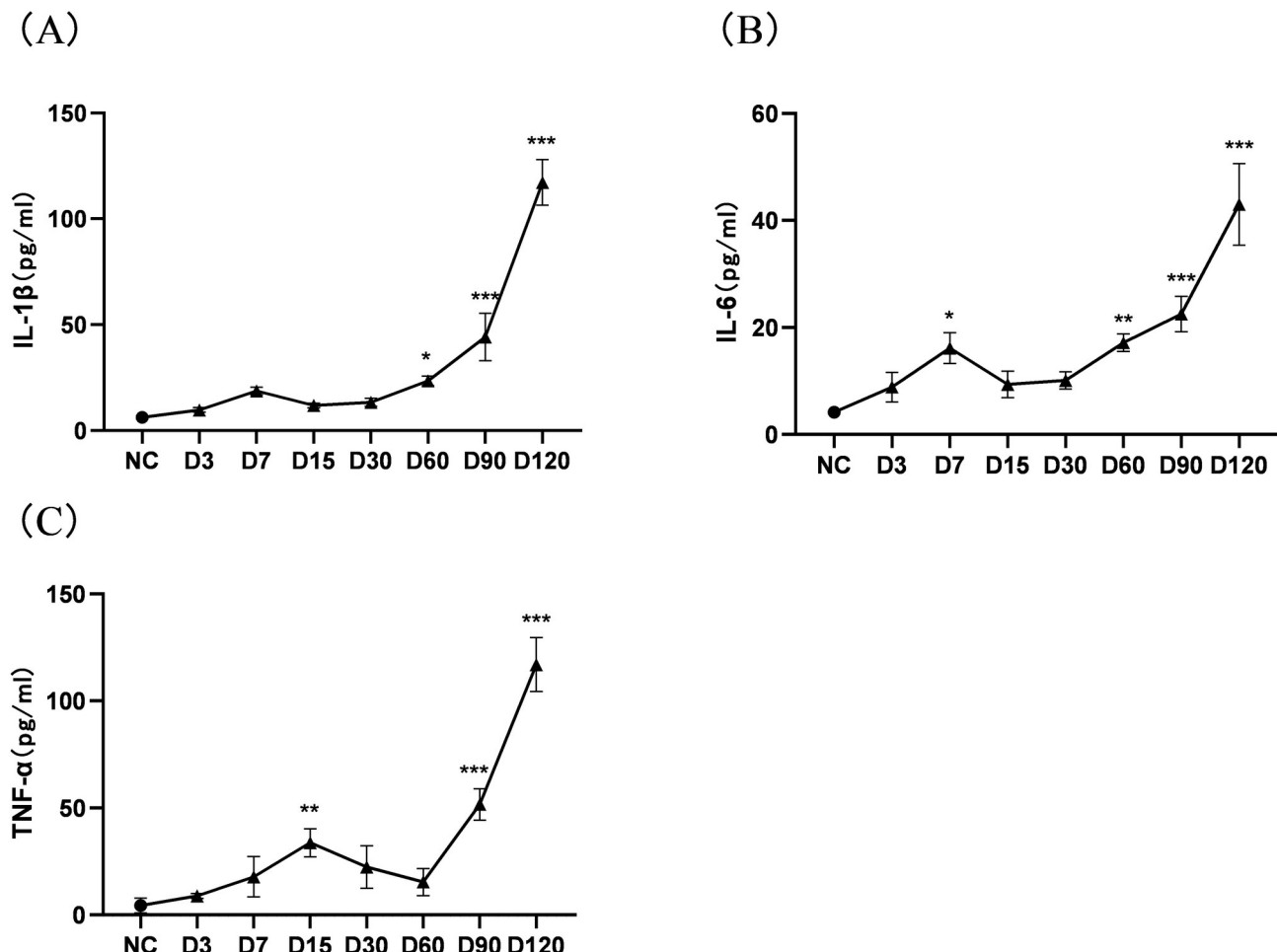

**Fig 6. Changes in the expression of inflammatory cytokines in BALF of mice exposed to continuous CS inhalation.** (A) The expression of IL-1 β; (B) The expression of IL-6; (C) The expression of TNF- α. Note: There is a difference compared to the control group, *: p<0.05; * *: P<0.01; * * *: P<0.001.

was a significant correlation between the level of sRAGE and FEV0.1, FEV0.1/FVC, neutrophil number and IL-1β levels in mouse BALF. The level of sRAGE was positively correlated with FEV0.1 (r = 0.955, P = 0.003) and FEV0.1/FVC (r = 0.969, P = 0.002). It was negatively correlated with neutrophil number (r = - 0.948, P = 0.004) and IL-1β level (r = - 0.869, P = 0.025) (Fig 8). The above results indicate that sRAGE expression was related to the degree of pulmonary function decline and inflammatory response.

## 4. Discussion

COPD is a disease characterized by chronic inflammation of the airways, lung parenchyma and pulmonary vasculature. Multiple pro-inflammatory pathways are involved in its pathogenesis. Among them, the AGEs-RAGE axis is considered to be an important pro-inflammatory pathway involved in this process. Tobacco smoking as the most important etiologic factor in the development of COPD has been found to be involved in the pathological formation of COPD by acting on the AGEs-RAGE axis. RAGE is a pattern recognition receptor with proinflammatory properties. The receptor is widely expressed in all organs and binds to its ligands to activate a variety of intracellular inflammatory signaling pathways [6]. RAGE is expressed at

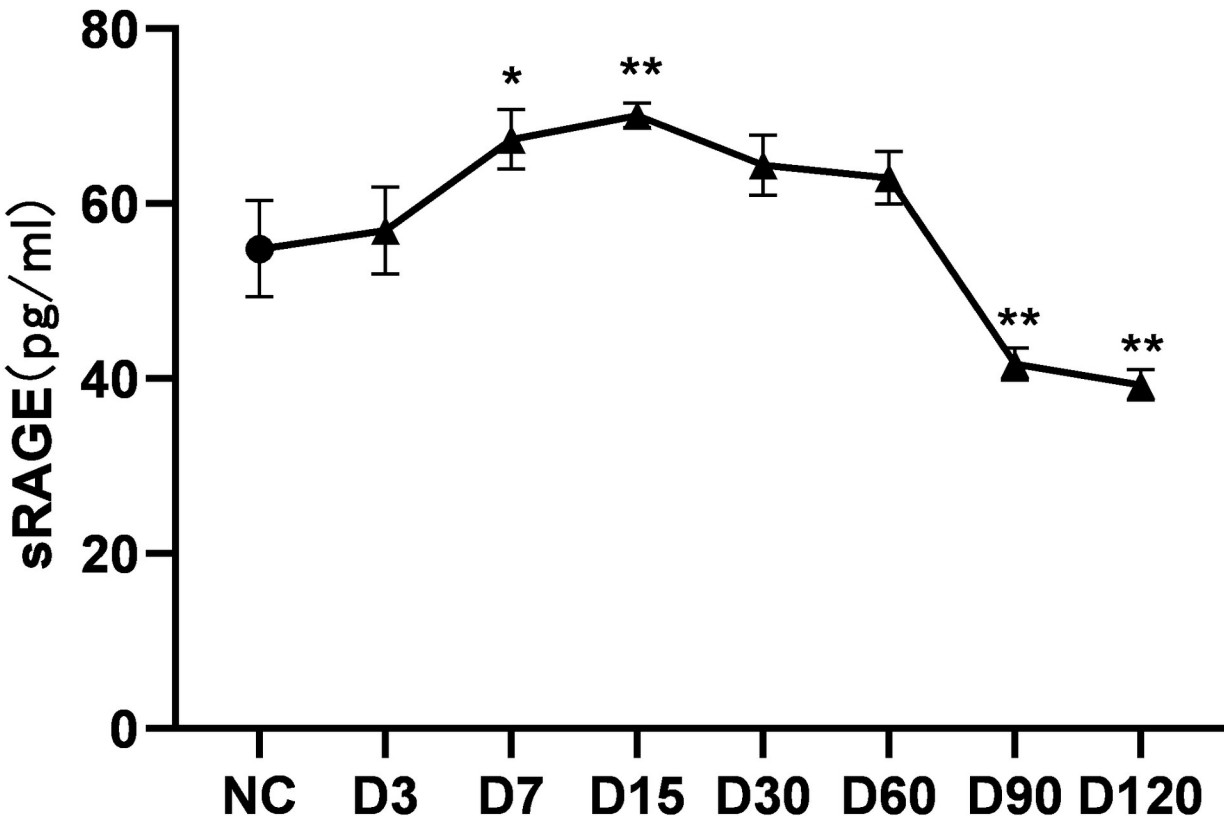

**Fig 7. Changes in sRAGE levels in BALF after 120 consecutive days of CS exposure.** Note: There is a difference compared to the control group, *: p<0.05; * *: P<0.01.

low basal levels in most tissues of healthy adults, whereas in the lungs, RAGE is expressed at high levels under normal physiological conditions [5]. sRAGE is a soluble form of RAGE that is produced by hydrolysis of the membrane-bound RAGE by proteases [12]. sRAGE is able to bind RAGE ligands and competitively inhibit the initiation of various RAGE-related pro-inflammatory signaling pathways. Therefore, sRAGE is also recognized as a "decoy receptor" with the ability to inhibit RAGE-associated inflammatory responses [7, 9].

sRAGE has demonstrated a correlation with COPD [3]. Several studies have shown that serum sRAGE levels are decreased in COPD patients. Cockayne et al., Iwamoto et al., Gopal et al., Pouwels et al., and Hoonhorst et al. all found that COPD patients had lower serum sRAGE levels compared to healthy smokers and non-smokers. Similarly, in Hoonhorst et al.'s study, it was found that patients with higher severity grades of COPD had lower serum sRAGE levels than those with lower grades [13–17].

Serum sRAGE levels have now been found to be significantly lower in COPD patients, but changes in sRAGE during the process from smoking to COPD formation remain controversial. Iwamoto et al. and Hoonhorst et al. have found there is no difference in serum sRAGE levels between healthy smokers and non-smokers [14, 15]. Biswat et al. have found that serum sRAGE levels in healthy smokers were significantly higher than those in non-smokers, and serum sRAGE levels were positively correlated with daily smoking quantity [18]. Pouwels et al., on the other hand, have designed experiments and concluded that smoking can rapidly reduces serum sRAGE levels [16]. In fact, it is very difficult to design an experiment with smoking as a single influencing factor in clinical practice and one of the major problems is that

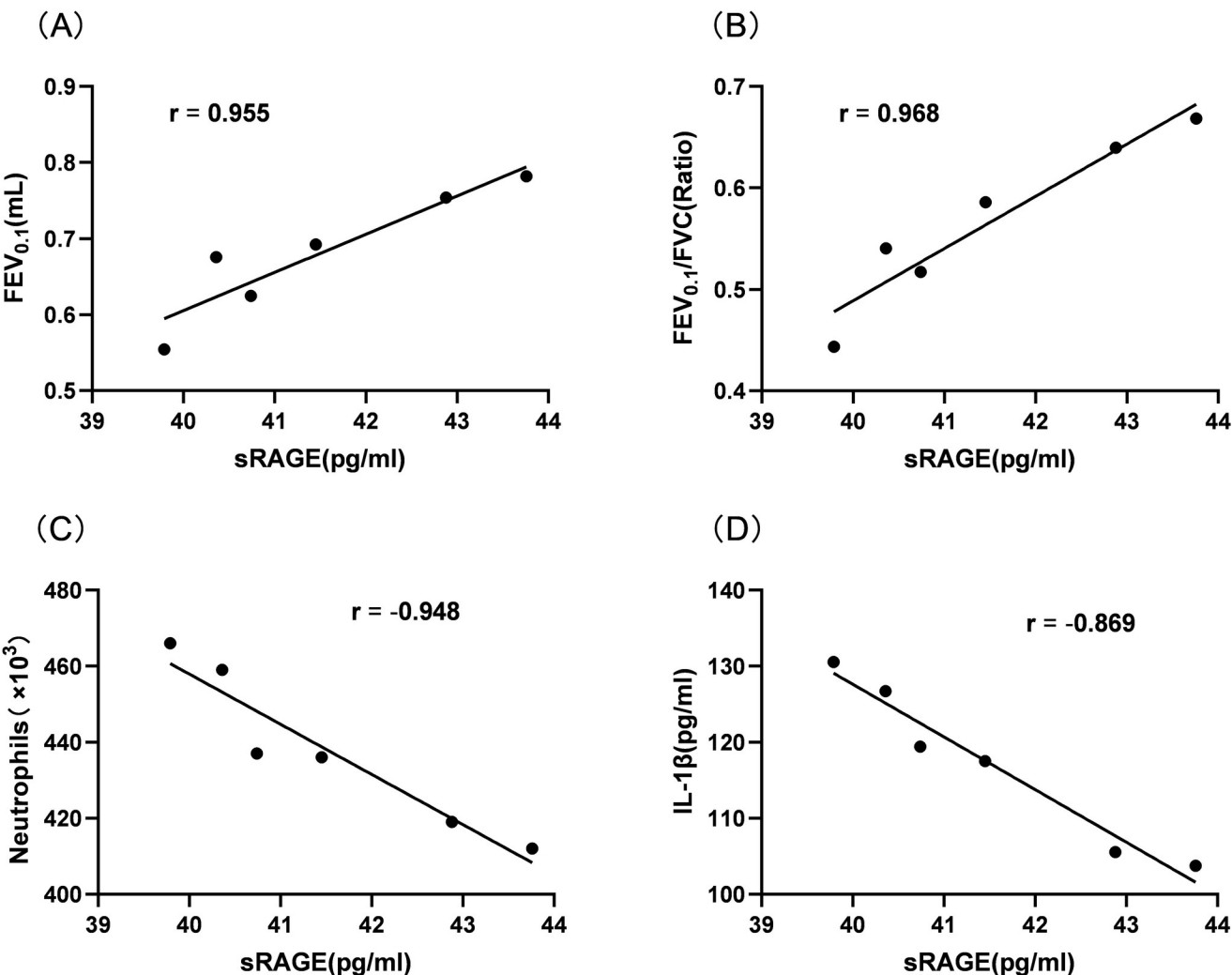

**Fig 8. Correlation between sRAGE levels and pulmonary dysfunction and inflammatory response in CS exposed 120 day group mice.** (A) The relationship between sRAGE and FEV0.1; (B) The relationship between sRAGE and FEV0.1/FVC; (C) The relationship between sRAGE and neutrophil count; (D) The relationship between sRAGE and IL-1 β.

the time and quantity of tobacco smoke exposure is difficult to control. More studies are needed to elucidate the impact of smoking on sRAGE levels and its potential mechanisms before sRAGE can be clinically applied as a biomarker for COPD.

To explore the true impact of smoking on sRAGE levels, we set a smoking exposure time gradient, constructed the COPD mouse model, and detected the changes of sRAGE expression levels in its BALF. The results of our study showed that sRAGE levels in the BALF of mice showed a dynamic change of first increasing and then decreasing during 120 days of exposure to cigarette smoke. At the early stages of tobacco smoke exposure (at two time points on day 7 and day 15), the expression level of sRAGE in BALF was upregulated, and the number of inflammatory cells and levels of inflammatory cytokines in BALF also increased to a certain extent. As tobacco smoke exposure continues, emphysema occurs and develops. The COPD mouse model was successfully constructed after 90 days of smoke exposure. The BALF sRAGE level was significantly lower than that of the control group at 90 days and decreased further with the extension of smoke exposure (120 days).

In addition, the results of this study suggest that low levels of sRAGE are associated with lung function impairment, and FEV 0.1, FEV0.1/FVC and PEF in the mouse model group of COPD were significantly reduced compared with those in the control group, and the central airway resistance was significantly increased. Further Pearson correlation analysis showed that sRAGE levels were negatively correlated with FEV0.1 and FEV0.1/FVC in COPD mice at 120 days of cigarette smoke exposure, suggesting that sRAGE is an important indicator of deterioration of lung function, which is consistent with the results of Iwamoto et al. [14]. We also found that sRAGE levels were inversely correlated with the severity of emphysema, with lower sRAGE levels being associated with higher levels of emphysema.

Our results show that the sRAGE content in BALF is up-regulated in the early stage and down-regulated after emphysema formation in response to tobacco smoke. We have tried to explain the reasons for this change. Smoking, as a harmful stimulus, accumulates AGEs in the body, especially because smoking accelerates oxidative stress, which is the most important factor in accelerating the formation and accumulation of AGEs [19]. In addition to AGEs, smoking upregulates the expression of various RAGE ligands, including HMGB1 and HSP70 [4, 20]. Serum AGEs were able to upregulate the expression of membrane-associated RAGE [21]. In the early stages of tobacco smoke exposure, the transient increase in sRAGE expression may be due to the accumulation of AGEs in the lungs stimulated by tobacco smoke, and the excess of AGEs leads to the upregulation of RAGE expression in the lungs, while sRAGE is produced by protease hydrolysis by membrane-bound RAGE [12], so the expression level of sRAGE decomposed by it is also upregulated. However, with the continuation of tobacco smoke exposure, the type 1 alveolar epithelial cells that mainly produce sRAGE are destroyed, and the consumption of sRAGE increases due to the binding of a large number of RAGE-related ligands to them, so the expression of sRAGE decreases during COPD formation [22]. We belive that the decline in sRAGE is inextricably linked to the deterioration of lung function.

Besides, this study investigated the changes in the inflammatory response of the lungs by measuring the number of inflammatory cells and the expression of inflammatory cytokines in BALF. Similar to the expression of sRAGE, we found that the inflammatory cytokines in BALF also showed early and late stage changes over the time course of smoke exposure. When mice smoked cigarette smoke for about 7 days, the levels of IL-1β, IL-6 and TNF-α in BALF increased transiently, suggesting that cigarette smoke exposure also caused lung inflammation in the early stage. This result is supported by previous study [23]. Even short-term smoking triggers an inflammatory response in the body, and the subsequent downregulation of inflammatory factor levels may be due to the body's anti-inflammatory counterregulation. In addition, Krüger et al. suggested that the temporary reduction of inflammation may be the result of the body's temporary habituation to smoke exposure [24]. After 90 days of cigarette smoke exposure, compared with the control group, the number of inflammatory cells in mouse BALF was significantly increased. Meanwhile, the levels of inflammatory cytokines IL-1β, IL-6 and TNF-α were significantly higher than those in the control group, this is consistent with previous studies [25]. The expression of potent inflammatory cytokines such as IL-1β and TNF-α increases with exposure to cigarette smoke and can play a key role in the development of airway inflammation and emphysema [2]. Harmful substances in cigarette smoke such as oxygen radicals, acrolein and formaldehyde can cause airway epithelial damage, and can also cause the aggregation of inflammatory cells, stimulate parietal cells to secrete IL-1β, IL-6 and other inflammatory cytokines, chemoattractant and activate neutrophils, release cytotoxic substances such as proteases, aggravate local inflammation and immune response of the airways, and eventually cause changes in the structure and function of small airways and lung tissues, leading to the occurrence and development of chronic bronchitis and emphysema [26].

## 5. Conclusion

In summary, this study established a mouse model of COPD through cigarette smoke induction, explored the dynamic changes of sRAGE and chronic inflammatory response during the onset of tobacco-induced COPD, and tried to explain the reasons for the level change of sRAGE in the gradual formation of COPD. In addition, we highlight the association between sRAGE and worsening of lung function, which enriches the theoretical basis for using sRAGE as a biomarker for COPD. However, since this study is descriptive, the causal relationship between different changes needs to be further explored.

## Supporting information

**S1 Dataset.**
(XLSX)

## Acknowledgments

To the Animal Laboratory and Basic Laboratory of Qingdao University for providing experimental venues. To the two anonymous reviewers for their help towards improving this work.

## Author Contributions

**Data curation:** Xiaohui Yang, Kai Huang.

**Methodology:** Fengyun Hao.

**Writing – review & editing:** Yue He, Hongyu Liang, Qiang Wang.

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
