## [Decision Letter · Decision Letter 0]

9 Sep 2024

PONE-D-24-20210Dynamic changes of lung sRAGE in mice with chronic obstructive pulmonary disease induced by cigarette smoke exposurePLOS ONE

Dear Dr. WANG,

Thank you for submitting your manuscript to PLOS ONE. After careful consideration, we feel that it has merit but does not fully meet PLOS ONE’s publication criteria as it currently stands. Therefore, we invite you to submit a revised version of the manuscript that addresses the points raised during the review process.

We look forward to receiving your revised manuscript.

Kind regards,

Misbahuddin Rafeeq

Academic Editor

PLOS ONE

Journal Requirements: When submitting your revision, we need you to address these additional requirements. 1. Please ensure that your manuscript meets PLOS ONE's style requirements, including those for file naming. The PLOS ONE style templates can be found at https://journals.plos.org/plosone/s/file?id=wjVg/PLOSOne_formatting_sample_main_body.pdf and https://journals.plos.org/plosone/s/file?id=ba62/PLOSOne_formatting_sample_title_authors_affiliations.pdf 2. To comply with PLOS ONE submissions requirements, in your Methods section, please provide additional information regarding the experiments involving animals and ensure you have included details on (1) methods of sacrifice, (2) methods of anesthesia and/or analgesia, and (3) efforts to alleviate suffering. 3. Thank you for stating the following in the Acknowledgments Section of your manuscript: "This work was supported by the Wu Jieping Special Fund for Clinical Research(320.6750.19092-6)." We note that you have provided funding information that is not currently declared in your Funding Statement. However, funding information should not appear in the Acknowledgments section or other areas of your manuscript. We will only publish funding information present in the Funding Statement section of the online submission form. Please remove any funding-related text from the manuscript and let us know how you would like to update your Funding Statement. Currently, your Funding Statement reads as follows: "This work was supported by the Wu Jieping Special Fund for Clinical Research(320.6750.19092-6).The author who received the funding: Wang QiangFunder：Wu Jiepinghttps://www.wjpmf.org.cn/No funders  play any role in the study design, data collection and analysis, decision to publish, or preparation of the manuscript" Please include your amended statements within your cover letter; we will change the online submission form on your behalf. 4. We note that your Data Availability Statement is currently as follows: All relevant data are within the manuscript and its Supporting Information files. Please confirm at this time whether or not your submission contains all raw data required to replicate the results of your study. Authors must share the “minimal data set” for their submission. PLOS defines the minimal data set to consist of the data required to replicate all study findings reported in the article, as well as related metadata and methods (https://journals.plos.org/plosone/s/data-availability#loc-minimal-data-set-definition). For example, authors should submit the following data: - The values behind the means, standard deviations and other measures reported;- The values used to build graphs;- The points extracted from images for analysis. Authors do not need to submit their entire data set if only a portion of the data was used in the reported study. If your submission does not contain these data, please either upload them as Supporting Information files or deposit them to a stable, public repository and provide us with the relevant URLs, DOIs, or accession numbers. For a list of recommended repositories, please see https://journals.plos.org/plosone/s/recommended-repositories. If there are ethical or legal restrictions on sharing a de-identified data set, please explain them in detail (e.g., data contain potentially sensitive information, data are owned by a third-party organization, etc.) and who has imposed them (e.g., an ethics committee). Please also provide contact information for a data access committee, ethics committee, or other institutional body to which data requests may be sent. If data are owned by a third party, please indicate how others may request data access. 5. When completing the data availability statement of the submission form, you indicated that you will make your data available on acceptance. We strongly recommend all authors decide on a data sharing plan before acceptance, as the process can be lengthy and hold up publication timelines. Please note that, though access restrictions are acceptable now, your entire data will need to be made freely accessible if your manuscript is accepted for publication. This policy applies to all data except where public deposition would breach compliance with the protocol approved by your research ethics board. If you are unable to adhere to our open data policy, please kindly revise your statement to explain your reasoning and we will seek the editor's input on an exemption. Please be assured that, once you have provided your new statement, the assessment of your exemption will not hold up the peer review process. 6. PLOS requires an ORCID iD for the corresponding author in Editorial Manager on papers submitted after December 6th, 2016. Please ensure that you have an ORCID iD and that it is validated in Editorial Manager. To do this, go to ‘Update my Information’ (in the upper left-hand corner of the main menu), and click on the Fetch/Validate link next to the ORCID field. This will take you to the ORCID site and allow you to create a new iD or authenticate a pre-existing iD in Editorial Manager. 7. Please review your reference list to ensure that it is complete and correct. If you have cited papers that have been retracted, please include the rationale for doing so in the manuscript text, or remove these references and replace them with relevant current references. Any changes to the reference list should be mentioned in the rebuttal letter that accompanies your revised manuscript. If you need to cite a retracted article, indicate the article’s retracted status in the References list and also include a citation and full reference for the retraction notice.

Reviewers' comments:

Reviewer's Responses to Questions

**Comments to the Author**

1. Is the manuscript technically sound, and do the data support the conclusions?

Reviewer #1: Yes

Reviewer #2: Yes

2. Has the statistical analysis been performed appropriately and rigorously? 

Reviewer #1: Yes

Reviewer #2: Yes

3. Have the authors made all data underlying the findings in their manuscript fully available?

Reviewer #1: Yes

Reviewer #2: Yes

4. Is the manuscript presented in an intelligible fashion and written in standard English?

Reviewer #1: Yes

Reviewer #2: Yes

5. Review Comments to the Author

Reviewer #1: - The paper is rigorously made and touched areas that needs further exploration and found to be a harbinger around the use of sRAGE as a pharmacological target. It worth’s to find a way to reach the wider scientific community.

Reviewer #2: In the article titled “Dynamic changes of lung sRAGE in mice with chronic obstructive pulmonary disease induced by cigarette smoke exposure” by Wang et. at., the authors measured the weight, lung function, the amount of MLI, MAN and PAA in the lungs, number of inflammatory cells, the amount of IL-1�, IL-6, and TNF-�, and the amount of sRAGES at 7 time points between 3 and 120 days, sacrificing 6 mice in the control and 6 mice at each time point. Mice exposed to cigarette smoke gained less weight, had decreased lung function over time and developed emphysema, had an increase in total number inflammatory cells, increased production of inflammatory cytokines, and initially higher sRAGES that decreased dramatically by the 120 day mark. Overall, the decrease in pulmonary function in the mice correlated with the increased inflammatory responses and the amount of sRAGES present. The strengths of the paper include n=6 at each time point, error bars present on all the graphs, the figure legends had a good description of figure it was describing, and the methods section was written so it could be easily be performed by another lab. The only weakness was that I found it confusing in Methods section regarding the 7 observation time points and n=6; I didn’t initially understand that 6 mice were sacrificed at each time point. Overall, I recommend that this article be published.

I only have 3 minor edits for the authors to consider.

1. It may make it more clear in the Methods section 1.3.1 Establishment of mouse model in the second sentence to state that 6 mice were sacrificed after each time point instead of only wring n=6 at the end of the sentence.

2. In the abstract, indicate what the acronym CS stands for so it is clear to your audience.

3. In the Discussion in the 11th line (7th sentence) RAGE is misspelled as RGAE.

6. PLOS authors have the option to publish the peer review history of their article (what does this mean?). If published, this will include your full peer review and any attached files.

Reviewer #1: No

Reviewer #2: No

---

## [Author Response · Author response to Decision Letter 0]

21 Oct 2024

Responding to editors:

1.

Editors' Comments to Author:

Response to comments:

We have carefully re-formatted the manuscript and changed the naming of the figures to suit PLOS ONE's style requirements. Ensure that it conforms to the style of your publication.

2.

Editors' Comments to Author:

To comply with PLOS ONE submissions requirements, in your Methods section, please provide additional information regarding the experiments involving animals and ensure you have included details on (1) methods of sacrifice, (2) methods of anesthesia and/or analgesia, and (3) efforts to alleviate suffering.

Response to comments:

We have amended the methods section of the manuscript for animal experiments to ensure that it includes detailed information on the methods of sacrifice, the methods of anesthesia and the efforts made to alleviate suffering.

3.

Editors' Comments to Author: 

Please remove any funding-related text from the manuscript and let us know how you would like to update your Funding Statement.

Response to comments:

We have removed any funding-related text from the manuscript. The funding statement is accurate and no changes need to be made.

4.

Editors' Comments to Author:

Please confirm at this time whether or not your submission contains all raw data required to replicate the results of your study. Authors must share the “minimal data set” for their submission.

Response to comments:

We've submitted all raw data required to replicate the results of our study and upload them as a Supporting Information file.

5.

Editors' Comments to Author:

When completing the data availability statement of the submission form, you indicated that you will make your data available on acceptance. We strongly recommend all authors decide on a data sharing plan before acceptance, as the process can be lengthy and hold up publication timelines. Please note that, though access restrictions are acceptable now, your entire data will need to be made freely accessible if your manuscript is accepted for publication. This policy applies to all data except where public deposition would breach compliance with the protocol approved by your research ethics board. If you are unable to adhere to our open data policy, please kindly revise your statement to explain your reasoning and we will seek the editor's input on an exemption. Please be assured that, once you have provided your new statement, the assessment of your exemption will not hold up the peer review process.

Response to comments:

We have modified our statement and decide to join the data sharing plan before acceptance.

6.

Editors' Comments to Author:

PLOS requires an ORCID iD for the corresponding author in Editorial Manager on papers submitted after December 6th, 2016. Please ensure that you have an ORCID iD and that it is validated in Editorial Manager.

Response to comments:

Thank you for pointing this out, the corresponding author has provided the ORCID iD and it has been validated in Editorial Manager.

7.

Editors' Comments to Author:

Response to comments:

We have checked all references in the manuscript to ensure that they are complete and accurate. For the references section, we made a modification by changing "曹君, 陈平, 杨悦, 欧阳若芸, 彭红. 烟雾暴露所致肺气肿小鼠模型的建立与评价. 中国实验动物学报. 2010;18: 278-282+366" to "Cao J, Chen P, Yang Y, Ouyang RY, Peng H. Establishment and Assessment of a Mouse Model of Cigarette Smoke-Induced Emphysema. Acta Laboratorium Animalis Scientia Sinica. 2010;18: 278-282+366".

Responding to reviewers:

1.

Reviewers ' Comments to Author:

It may make it more clear in the Methods section 1.3.1 Establishment of mouse model in the second sentence to state that 6 mice were sacrificed after each time point instead of only wring n=6 at the end of the sentence.

Response to comments:

We removed the statement here and recharacterized our grouping in clearer terms.

2.

Reviewers ' Comments to Author:

In the abstract, indicate what the acronym CS stands for so it is clear to your audience.

Response to comments:

In the abstract of our manuscript, we indicated what the acronym CS stands for.

3.

Reviewers ' Comments to Author:

In the Discussion in the 11th line (7th sentence) RAGE is misspelled as RGAE.

Response to comments:

Thank you for your careful review, we have corrected the typos.

4.

Reviewers ' Comments to Author:

Abstract TNF-αin should be TNF-α in.

Response to comments:

We have fixed the error here.

5.

Reviewers ' Comments to Author:

Clinical Research(320.6750.19092-6) should be Clinical Research (320.6750.19092-6).

Response to comments:

We have fixed the error here.

6.

Reviewers ' Comments to Author:

You used different line spaces at “These results showed that cigarette smoke exposure could induce obstructive ventilatory dysfunction characterized by emphysema in mice, and lung function decreased with prolonged…”.

Response to comments:

We have corrected the formatting error here.

7.

Reviewers ' Comments to Author:

15th day should be 15th.

Response to comments:

We have fixed the error here.

8.

Reviewers ' Comments to Author:

Use Pearson correlation … should be Pearson correlation analysis was used …

Response to comments:

We have fixed the error here.

9.

Reviewers ' Comments to Author:

Tobacco smoking as an most important etiologic… should be as the most.

Response to comments:

We have fixed the error here.

10.

Reviewers ' Comments to Author:

Some of the discussion paragraphs are too long and cumbersome, it would be better if fragmented in a certain way and made concise as much as possible.

Response to comments:

We have streamlined some of the discussion sections and added new subsections to make them easier to read.

---

## [Editor Report · Decision Letter 1]

1 Nov 2024

Dynamic changes of lung sRAGE in mice with chronic obstructive pulmonary disease induced by cigarette smoke exposure

PONE-D-24-20210R1

Dear Dr. Qiang Wang,

We’re pleased to inform you that your manuscript has been judged scientifically suitable for publication and will be formally accepted for publication once it meets all outstanding technical requirements.

Kind regards,

Misbahuddin Rafeeq

Academic Editor

PLOS ONE

---

## [Editor Report · Acceptance letter]

15 Nov 2024

PONE-D-24-20210R1 

PLOS ONE

Dear Dr. WANG, 

I'm pleased to inform you that your manuscript has been deemed suitable for publication in PLOS ONE. Congratulations! Your manuscript is now being handed over to our production team.

Kind regards, 

on behalf of

Dr. Misbahuddin Rafeeq 

Academic Editor

PLOS ONE